# Impact of a decade of successful antiretroviral therapy initiated at HIV-1 seroconversion on blood and rectal reservoirs

Eva Malatinkova[1†], Ward De Spiegelaere[1†], Pawel Bonczkowski[1], Maja Kiselinova[1], Karen Vervisch[1], Wim Trypsteen[1], Margaret Johnson[2], Chris Verhofstede[3], Danny de Looze[4], Charles Murray[5], Sabine Kinloch-de Loes[2†], Linos Vandekerckhove[1*†]

[1]HIV Translational Research Unit, Department of Internal Medicine, Faculty of Medicine and Health Sciences, Ghent University and Ghent University Hospital, Ghent, Belgium; [2]Division of Infection and Immunity, Royal Free Hospital, London, United Kingdom; [3]AIDS Reference Laboratory, Department of Clinical Chemistry, Microbiology and Immunology, Ghent University, Ghent, Belgium; [4]Department of Gastroenterology, Ghent University Hospital, Ghent, Belgium; [5]Department of Gastroenterology, Royal Free Hospital, London, United Kingdom

*For correspondence: Linos. Vandekerckhove@ugent.be

†These authors contributed equally to this work

Competing interests: The authors declare that no competing interests exist.

**Abstract** Persistent reservoirs remain the major obstacles to achieve an HIV-1 cure. Prolonged early antiretroviral therapy (ART) may reduce the extent of reservoirs and allow for virological control after ART discontinuation. We compared HIV-1 reservoirs in a cross-sectional study using polymerase chain reaction-based techniques in blood and tissue of early-treated seroconverters, late-treated patients, ART-naïve seroconverters, and long-term non-progressors (LTNPs) who have spontaneous virological control without treatment. A decade of early ART reduced the total and integrated HIV-1 DNA levels compared with later treatment initiation, but not reaching the low levels found in LTNPs. Total HIV-1 DNA in rectal biopsies did not differ between cohorts. Importantly, lower viral transcription (HIV-1 unspliced RNA) and enhanced immune preservation (CD4/CD8), reminiscent of LTNPs, were found in early compared to late-treated patients. This suggests that early treatment is associated with some immunovirological features of LTNPs that may improve the outcome of future interventions aimed at a functional cure.

## Introduction

A reservoir of long-lived latently HIV-1 infected cells is established early in the course of the infection. It persists despite suppressed viremia in patients undergoing effective antiretroviral therapy (ART) and fuels viral rebound upon treatment discontinuation (*Wong et al., 1997*; *Finzi et al., 1997*; *Chun et al., 1997*; *Finzi et al., 1999*; *Fernandez et al., 2005*; *Alexaki et al., 2008*). Not only is this reservoir present in blood, but also in tissues such as lymphoid organs, the gut and potentially the central nervous system (*Chun et al., 2008*; *Sturdevant et al., 2015*; *Bednar et al., 2015*).

The mechanisms underlying HIV-1 persistence have not been fully elucidated. Although an initial decay of these reservoirs is observed after ART intervention, it is assumed that replenishment may occur through clonal proliferation of infected CD4 T cells during ART (*Chomont et al., 2009*; *Josefsson et al., 2013*; *Maldarelli et al., 2014*; *Murray et al., 2014*) or through residual virus production despite suppressive ART (*Chun et al., 2008*; *Buzon et al., 2010*; *Hatano et al., 2013a*)

**eLife digest** Many people with HIV infections are able to live relatively normal lives thanks to major advances in drug therapies. A cure, however, remains elusive. One reason for this is that the virus can hide in certain types of human cells, where it is protected from the immune system and the effects of "antiretroviral" drugs. This creates reservoirs of virus particles in the body that can quickly multiply and spread if treatment stops.

Some people who become infected with HIV are able to contain the virus without the help of drug treatments. These individuals – known as long-term non-progressors – do not become ill and only have low numbers of HIV particles in reservoirs. People who receive treatment early in the course of an HIV infection also have fewer viruses in reservoirs and are less likely to develop severe illness. Therefore, it might be possible to develop a "functional" cure that may not completely eliminate the virus from the body, but would prevent illness and allow the individuals to eventually stop taking antiretroviral drugs.

Now, Malatinkova, De Spiegelaere et al. studied samples from 84 patients with HIV-1 to find how much effect an early start to treatment has on the amount of the virus in reservoirs. People who started treatment soon after infection had lower levels of HIV-1 in their blood than people who started treatment later (even after 10 years of treatment). However, patients that started treatment early had higher levels of HIV-1 in the blood than the patients who were long-term non-progressors. All the patients had similar levels of HIV-1 in tissue samples taken from the rectum, regardless of when they started treatment.

The experiments suggest that HIV-1 reservoirs form very soon after infection. Malatinkova, De Spiegelaere et al. found that in addition to reducing reservoirs of HIV-1, an early start to drug treatment reduced the ability of the virus to make copies of its genetic code. People who started treatment earlier also had healthier immune cells. Together, the experiments support the benefits of starting drug treatments as soon as possible after a person is infected with HIV-1. It is important to further characterize thoroughly the viral reservoir in patients with limited HIV-1 reservoirs and to look for other immune factors involved in virus control, in the search for a functional cure of HIV.

possibly in sanctuary sites where ART penetration is suboptimal (*Yukl et al., 2010*; *Fletcher et al., 2014*).

Low levels of viral reservoirs have been associated with an absence of viral rebound after treatment discontinuation in several case reports and the Visconti cohort, suggesting the possibility of post-treatment virological control even in the presence of viral reservoirs (*Salgado et al., 2011*; *Van Gulck et al., 2012*; *Saez-Cirion et al., 2013*; Kinloch-de Loes et al., personal communication). Achieving such a long-term control of HIV-1 replication in the absence of ART is widely defined as a functional cure (*Saag and Deeks, 2010*; *Fauci and Folkers, 2009*). A low saturation of viral reservoirs facilitated by early treatment initiation might be a necessary condition, although not in itself sufficient for post-treatment virological control (*Saez-Cirion et al., 2013*; *Van Gulck et al., 2012*). Recent evidence indicates that the interplay between virological and immunological parameters is likely to be fundamental to achieve this goal (*Cellerai et al., 2011*). A sustained remission from viremia rebound seems a more realistic prospect in terms of HIV-1 cure research in the short term (*Katlama et al., 2013*).

Early treatment initiation with ART will likely become the standard clinical practice in HIV care. This is supported by the recent outcome of the first large-scale international 'Strategic Timing of AntiRetroviral Treatment' (START) study, showing a considerably lower risk of developing AIDS and other serious conditions when compared to later treatment initiation (*INSIGHT START Study Group, 2015*). Interestingly, early treatment initiation during HIV-1 seroconversion is also the most effective intervention to limit the extent of viral reservoirs (*Ananworanich et al., 2012*; *Hoen et al., 2007*; *Hocqueloux et al., 2013*; *Ananworanich et al., 2015*). Very low or even undetectable HIV-1 DNA has been described when treatment is initiated during the very early stages of primary HIV-1 infection (PHI) (*Ananworanich et al., 2012*; *Laanani et al., 2015*). In addition, a lower level of HIV-1 transcription has been described in ART-treated patients who initiated treatment during seroconvertion (*Schmid et al., 2010*).

Elite controllers and long-term non-progressors (LTNPs) represent an important group as a comparator. These HIV-1 infected individuals display low or undetectable blood reservoirs, and are able to control viremia over the long-term with limited CD4 T cell loss in the absence of treatment. Consequently, LTNPs have been extensively studied in an attempt to unravel the underlying mechanisms of spontaneous virological control (*Autran et al., 2011*; *Deeks and Walker, 2007*). Although their viral reservoirs have been shown to be low, replication-competent viruses can still be found in these individuals (*Blankson et al., 2007*; *Buzon et al., 2014*). LTNPs display strong HIV-1-specific T cells responses with polyfunctionality, thereby suggesting a role of T cell immunity in viremia control (*Cellerai et al., 2011*).

In the present study, we have assessed whether undetectable or low levels of reservoirs in blood and tissue could be achieved with very prolonged therapy initiated at PHI or during chronic infection using newly-developed polymerase chain reaction (PCR)-based virological assays for in-depth measurement of the size of the HIV-1 reservoir in blood (total and integrated HIV-1 DNA) and its dynamics (episomal 2-long terminal repeat (LTR) circles and cell-associated unspliced RNA [usRNA]) as well as total HIV-1 DNA burden in the rectal mucosa. These patients were compared to LTNPs and to untreated seroconverters.

We have analyzed whether 1) a decade of ART or an LTNP status was associated with the absence of detectable HIV-1 DNA in the blood and rectal mucosa; 2) long-term treated seroconverters could reach levels of virological reservoirs, residual replication, and transcription comparable to those of LTNPs; 3) a similar period of aviremia with ART initiation during the chronic phase of HIV-1 infection could achieve levels of reservoirs, residual replication, and transcription comparable to long-term treated seroconverters; 4) immune reconstitution, as measured by CD4/CD8 ratio, was enhanced with early treatment intervention; and 5) a correlation was present between the various virological and immunological parameters used in this study.

## Results

### Study participant characteristics

Eighty-four patients were included in this cross-sectional study from four different cohorts: patients who had undergone a decade of successful ART, initiated either during seroconversion (SRCV on ART; n = 25) or during the chronic phase of the infection (Chronic ART, n = 32), LTNPs (n = 17), and recently infected ART-naïve seroconverters (Recent SRCV; n = 10) (*Figure 1*; *Table 1*). The CD4 nadir was significantly different between each of the cohorts (p < 0.001) except between SRCV on ART and Recent SRCV (p = 0.623). The patients were sampled at a single time point (blood and rectal biopsies) to perform PCR-based assays and characterize viral reservoirs and its dynamics (total and integrated HIV-1 DNA, 2-LTR circles, and HIV-1 usRNA).

### Total and integrated HIV-1 DNA in blood is reduced after a decade of ART with early treatment initiation, but does not reach the low levels found in LTNPs

Total HIV-1 DNA represents the most commonly used virological marker for the assessment of the size of the proviral HIV-1 reservoir and is predictive of viral rebound when tested at the time of treatment interruption in early-treated patients (*Williams et al., 2014*). We have assessed the impact of the temporal treatment initiation (during early versus chronic infection) in the context of very prolonged ART treatment (e.g. a decade) on this marker and compared results to those of LTNPs and acute seroconverters before ART initiation to assess the size of HIV-1 DNA using digital PCR.

All patient cohorts had detectable levels of total HIV-1 DNA in peripheral blood mononuclear cells (PBMCs). Differences in the reservoir size in terms of total HIV-1 DNA were observed between patient cohorts. Median total HIV-1 DNA was: 92 (interquartile range (IQR): 9.8–127.7), 48 (IQR: 5.4–56.5), 137 (IQR: 8.6–219.2) and 1901.3 (IQR: 602.4–4786.9) copies (c)/$10^6$ PBMCs in SRCV on ART, LTNPs, Chronic ART, and Recent SRCV, respectively. Lower total HIV-1 DNA was detected in the SRCV on ART compared to the Chronic ART cohort (p = 0.041; *Figure 2A*). The LTNP cohort showed the lowest total HIV-1 DNA levels when compared to SRCV on ART (p = 0.015) and other patient cohorts (p < 0.001; *Figure 2A*). These results demonstrate that, the total HIV-1 DNA remains

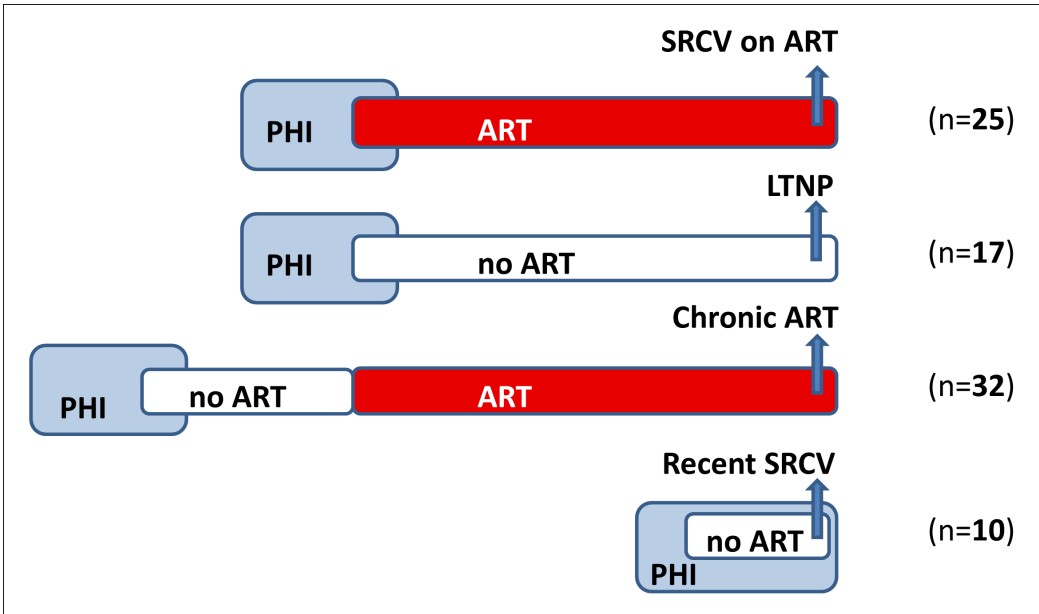

**Figure 1.** Patient cohorts in the cross-sectional study. SRCV on antiretroviral therapy (ART): patient cohort with ART initiated at the time of HIV-1 seroconversion; LTNP: long-term non-progressors; Chronic ART: patients with ART initiated during the chronic phase of HIV-1 infection; Recent SRCV: recent ART-naïve seroconverters. In total, 84 patients were included in this study, 25 in SRCV on ART, 17 LTNPs, 32 Chronic ART patients, and 10 Recent SRCV. Blue arrows represent time of sampling. PHI: primary HIV-1 infection.

**Table 1.** Clinical and laboratory characteristics of the four patient cohorts.

| | Value for cohort* | | | |
| --- | --- | --- | --- | --- |
| | **SRCV on ART** | **LTNP** | **Chronic ART** | **Recent SRCV** |
| | **n = 25** | **n = 17** | **n = 32** | **n = 10** |
| **Clinical characteristics** | | | | |
| Age (years) | 44 (34–53) | 49 (31–51) | 48 (31–53) | 39 (30–46) |
| Number of females (%) | 0 (0) | 7 (41.2) | 5 (15.6) | 1 (10) |
| Total ART duration (years) | 10.8 (4.2–11.9) | 0 | 9.8 (4.9–14.7) | 0 |
| Viremia zenith (log$_{10}$HIV-1 c/ml) | 5.5 (2.4–5.9) | 2.5 (1.6–2.8) | 4.9 (1.9–5.5) | 6.2 (5.2–6.4) |
| CD4 count, nadir (cells/mm$^3$) | 390 (107–466) | 624 (507–693) | 155 (0–266) | 440 (284–495) |
| CD4 count at sampling (cells/mm$^3$) | 714 (476–977) | 793 (414–1010) | 624.5 (172–889) | 440 (284–604) |
| CD4/CD8 ratio | 1.10 (0.52–1.35) | 0.91 (0.36–1.47) | 0.74 (0.23–0.93) | 0.62 (0.36–0.94) |
| **Virological markers** | | | | |
| Total HIV-1 DNA (c/10$^6$ PBMCs) | 92 (9.8–127.7) | 48 (5.4–56.5) | 137 (8.6–219.2) | 1901.3 (602.4–4786.9) |
| Integrated HIV-1 DNA (c/10$^6$ PBMCs) | 211.3 (0–589.3) | 28.2 (0–158.4) | 586.7 (131.6–918.2) | 1802.7 (367.9–5890.6) |
| HIV-1 usRNA (c/10$^6$ PBMCs) | 1.6 (0–3.7) | 0.4 (0–3.51) | 6.1 (0–10.1) | 15.5 (0.9–100.6) |
| 2-LTR circles (c/10$^6$ PBMCs) | 2.1 (0–4.3) | 0.8 (0–2.7) | 1.3 (0–2.2) | 13.3 (5.1–21.6) |
| Total HIV-1 DNA (c/10$^6$ cells) in rectal biopsies | 27.2 (22.2–61.7) | 21.3 (16.7–34.5) | 35.1 (16–77.5) | |

*Values are reported as median (IQR); c: copies; PBMCs: peripheral blood mononuclear cells; usRNA: unspliced RNA; ART: antiretroviral therapy; SRCV on ART: early treated seroconverters; LTNP: long-term non-progressors; Chronic ART: late treated patients during chronic HIV-1 infection; Recent SRCV: acute ART-naïve seroconverters; LTR: Long terminal repeat.

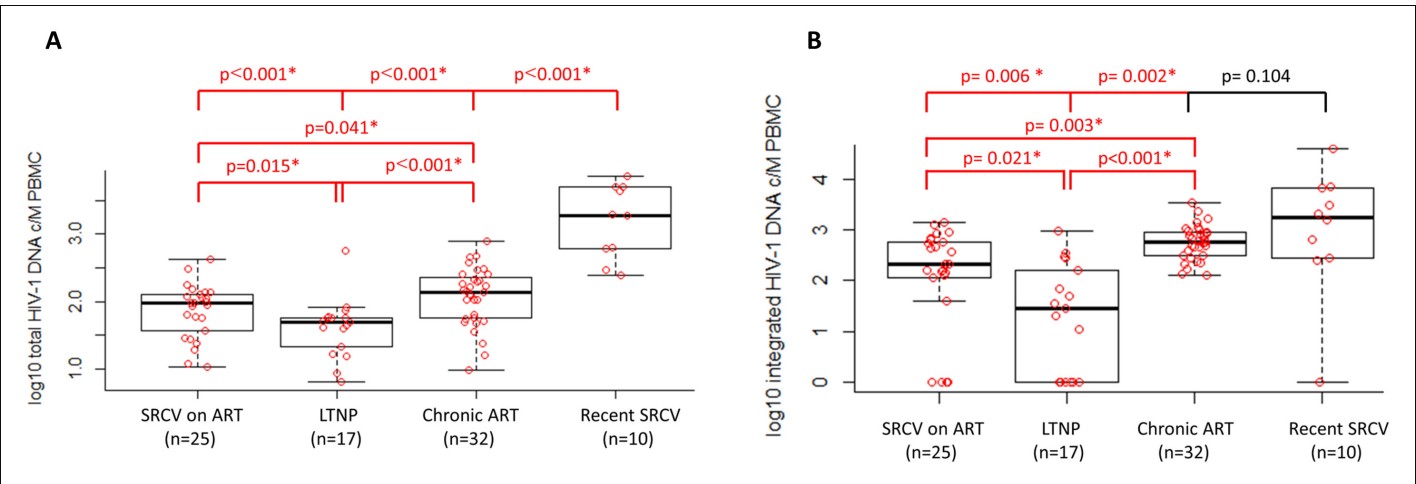

**Figure 2.** Total HIV-1 DNA (**A**) and integrated HIV-1 DNA (**B**) in four patient cohorts. Data is shown as $\log_{10}$ copies/million (c/M) PBMCs and significant p-values are indicated by *. Differences between the cohorts were determined by Wilcoxon Signed Rank test.

detectable in all patients even in the setting of effective early and prolonged ART, although at lower levels than in patients with later ART initiation, but not reaching the levels found in LTNPs.

In addition to total HIV-1 DNA levels, we also measured integrated HIV-1 DNA levels. This marker is not biased by the presence of variable quantities of unintegrated HIV-1 DNA produced after reverse transcription of newly infecting HIV-1 or dead-end DNA products of failed integration such as 1- and 2-LTR circles. Some authors have suggested that integrated HIV-1 DNA represents a better surrogate marker of viral burden, especially in patients off ART in whom HIV-1 replication might be ongoing (*Graf et al., 2011*).

Median integrated HIV-1 DNA levels were: 211.3 (IQR: 0–589.3), 28.2 (IQR: 0–158.4), 586.7 (IQR: 131.6–918.2), and 1802.7 (IQR: 367.9–5890.6) $c/10^6$ PBMCs for SRCV on ART, LTNPs, Chronic ART, and Recent SRCV, respectively. A lower level of integrated HIV-1 DNA was found in SRCV on ART compared to the Chronic ART cohort (p = 0.003) and in LTNPs compared to both the SRCV on ART (p = 0.021) and Chronic ART cohorts (p < 0.001; *Figure 2B*). These results confirm the low levels of integrated DNA found in LTNPs as described previously (*Graf et al., 2011*). Interestingly, levels of integrated HIV-1 DNA were not significantly different between the ART-naïve Recent SRCV and the Chronic ART cohort (p = 0.104). The Recent SRCV cohort displayed higher integrated HIV-1 DNA levels compared to SRCV on ART (p = 0.006) and LTNPs (p = 0.002; *Figure 2B*).

Of note, the absolute values derived from total and integrated HIV-1 DNA measurements in this study cannot be directly compared with each other. This is due to the difference in quantification methods, the absolute quantification of integrated HIV-1 DNA being corrected by using an integration standard as calibrator as previously described (*Liszewski et al., 2009*; *De Spiegelaere et al., 2014*). This calibrator does not alter the relative differences of integrated HIV-1 DNA between patient samples, but may bias the absolute quantitative outcome. In contrast, total HIV-1 DNA is reported by direct absolute quantification and represents a better estimate of the amount of HIV-1 DNA $c/10^6$ cells present in patients.

Due to the lower limit of detection compared to that of the total HIV-1 DNA assay, integrated HIV-1 DNA was undetectable in 12 patients; 5 were SRCV on ART, 6 LTNPs, and 1 Recent SRCV.

## Levels of episomal 2-LTR circles in blood are low in ART-treated patients and LTNPs but high in recent ART-naïve seroconverters

Residual viral replication is one of the likely mechanisms through which the viral reservoir is replenished, even with effective ART (*Hong and Mellors, 2015*). Markers that reflect such a phenomenon provide insight into the reservoir dynamics. Episomal 2-LTR circles can be used as a marker of viral replication and have been shown to be labile end-products of failed proviral integrations (*Sharkey et al., 2005*; *2011*). They are likely to be particularly elevated in acutely infected patients

because of the intense level of viral replication. Median levels of 2-LTR circles in the various cohorts were: 2.1 (IQR: 0–4.3), 0.77 (IQR: 0–2.7), 1.3 (IQR: 0–2.2) and 13.3 (IQR: 5.1–21.6) c/$10^6$ PBMCs in SRCV on ART, LTNPs, Chronic ART, and Recent SRCV, respectively. As expected, 2-LTR levels were significantly higher in the Recent SCRV compared to the other cohorts (p $\leq$ 0.002; *Figure 3A*), but did not differ between SRCV on ART and Chronic ART (p = 0.259) or LTNPs (p = 0.595) or between the Chronic ART and LTNP cohorts (p = 0.743; *Figure 3A*). Of note, 2-LTR circles were undetectable in a number of patients. They were detected in 17/25 (68%) of SRCV on ART, 13/17 (76%) of LTNPs, 25/32 (78%) of late-treated patients and in all recent ART-naïve seroconverters 10/10 (100%). The absence of detection in about one-quarter of patients on ART in contrast to the Recent SRCV cohort suggests that residual replication is substantially halted with ART.

## Cell-associated HIV-1 usRNA levels in blood are lower in early-treated seroconverters than in late-treated patients but not different compared to LTNPs

Levels of cell-associated HIV-1 usRNA are associated with recent HIV-1 transcriptional activity and indicate an active proviral reservoir (*Pasternak et al., 2013*). Median levels were: 1.6 (IQR: 0–3.7), 0.4 (IQR: 0–3.5), 6.1 (IQR: 0–10.1) and 15.5 (IQR: 1–100.6) c/$10^6$ PBMCs in SRCV on ART, LTNPs, Chronic ART, and Recent SRCV, respectively. The patients with the highest levels of HIV-1 usRNA were found in the Recent SRCV cohort, indicating that this cohort is not similar to the others, but

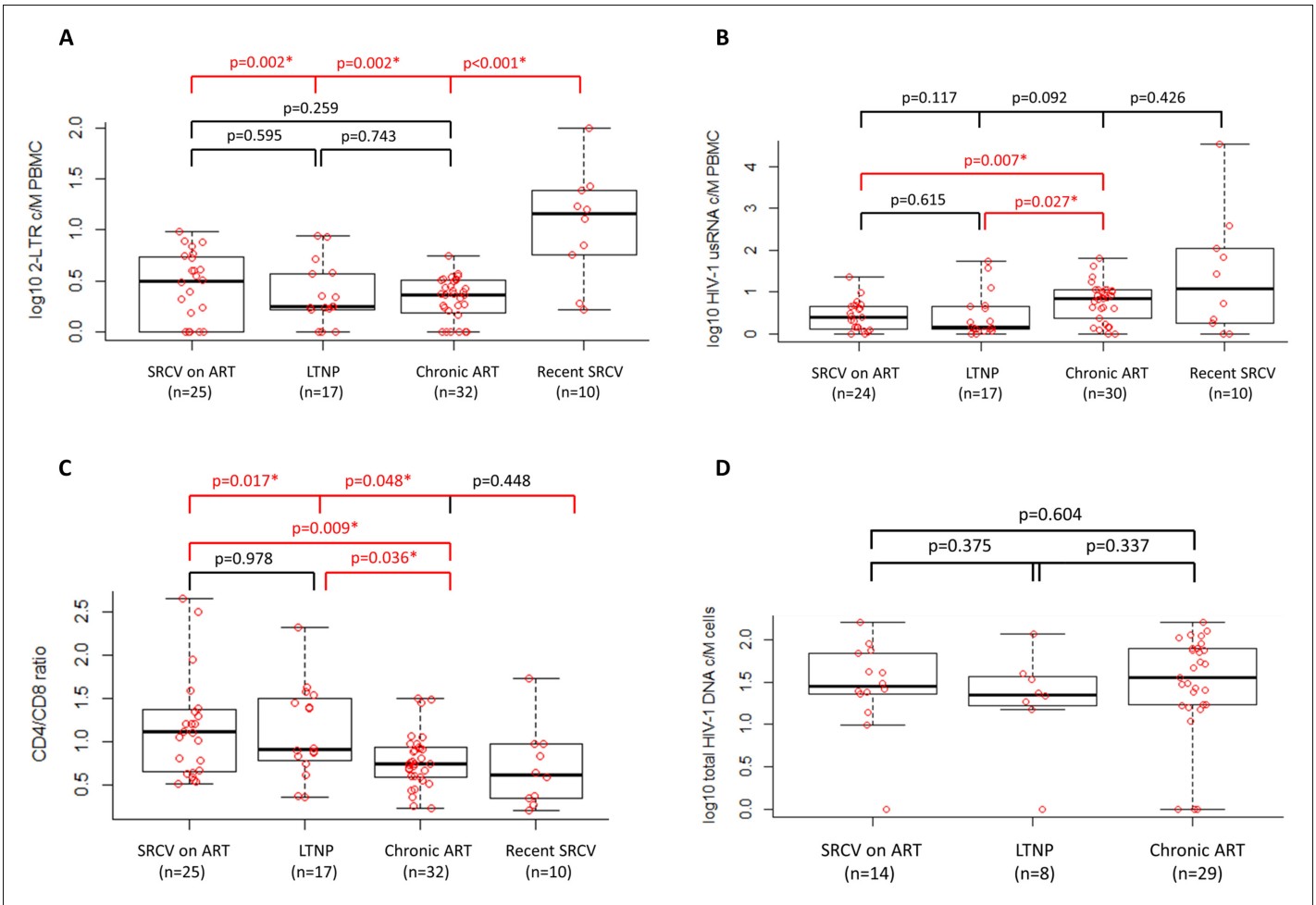

**Figure 3.** 2-LTR circles (A), cell-associated HIV-1 usRNA (B) and CD4/CD8 ratio (C) in four patient cohorts. Total HIV-1 DNA in rectal biopsies (D) in three patient cohorts (SRCV on ART, LTNP, and Chronic ART). Data is shown as $log_{10}$ copies/million (c/M) PBMCs (A,B), ratio (C) or $log_{10}$ c/M cells in rectal biopsies (D) and significant p-values are indicated by *. Differences between the cohorts were determined by Wilcoxon Signed Rank test.

this study was underpowered to reach statistical significance between the Recent SRCV and the other cohorts. Of note, HIV-1 usRNA was not detected in 8/84 samples (10%, two samples in each patient cohort). Three samples (one in SRCV on ART and two in Chronic ART) were excluded from the final analysis due to missing values for the reference genes. Higher levels of HIV-1 usRNA were detected in the Chronic ART cohort compared to the SRCV on ART (p = 0.007) and LTNPs (p = 0.027; *Figure 3B*). The SRCV on ART cohort was not significantly different from LTNPs based on HIV-1 usRNA levels (p = 0.615; *Figure 3B*).

### A higher CD4/CD8 ratio is present in early-treated seroconverters and LTNPs compared to late-treated patients

Not only have we used the CD4/CD8 ratio as a measure of immune preservation/reconstitution in terms of T cell count but also as an indirect marker of residual immune activation as shown recently (*Serrano-Villar et al., 2014*). The CD4/CD8 ratio was higher in SRCV on ART (median = 1.10, IQR: 0.52–1.35) compared to the cohorts of Chronic ART (median = 0.74, IQR: 0.23–0.93), (p = 0.009) and Recent SRCV (median = 0.62, IQR: 0.36–0.94), (p = 0.017), and was comparable to that of LTNPs (median = 0.91, IQR: 0.36–1.47), (p = 0.978; *Figure 3C*). The LTNP cohort had a higher CD4/CD8 ratio compared to late-treated patients (p = 0.036) and Recent SRCV (p = 0.048; *Figure 3C*). Of note, CD4 T cell counts at sampling did not differ significantly between the cohorts of SRCV on ART (median = 714 cells/mm$^3$, IQR: 476–977) and Chronic ART (median = 625 cells/mm$^3$, IQR: 172–889), (p = 0.066).

### Total HIV-1 DNA levels in rectal biopsies are not different in early-treated seroconverters, late-treated patients, and LTNPs

Conflicting results have been published regarding the impact of ART on the HIV-1 DNA reservoir in the gut compartment when using rectal biopsies (*Yukl et al., 2010*; *Chun et al., 2008*; *Anton et al., 2003*). It remains unclear whether HIV-1 DNA decays more substantially after a decade of ART compared to a shorter intervention and whether some LTNPs may have undetectable levels in rectal biopsies. We have measured total HIV-1 DNA in rectal biopsies from 51 patients who had consented to sampling: 14 SRCV on ART, 8 LTNPs, and 29 Chronic ART patients. Five patients had undetectable total HIV-1 DNA; one SRCV on ART, one LTNP, and three Chronic ART patients. Median HIV-1 DNA levels were: 27.2 (IQR: 22.2–61.7), 21.3 (IQR: 16.7–34.5), and 35.1 (IQR: 16–77.5) c/10$^6$ cells in SRCV on ART, LTNP, and Chronic ART, respectively. No difference was found between SRCV on ART and Chronic ART (p = 0.604) or LTNPs (p = 0.375; *Figure 3D*) or between the Chronic ART cohort and the LTNPs (p = 0.337; *Figure 3D*). The median number of cells assayed per patient was 125,390 (IQR: 101,308–168,156). Of note, we did not find any correlation in terms of total HIV-1 DNA levels between the blood (PBMCs) and the gut mucosa (rectal biopsies) compartments (R$^2$ = 0, p = 0.919; *Figure 4A*).

### Correlation between viroimmunological markers

In order to assess whether we could observe correlations between the various viroimmunological markers used in this study, we have performed a linear regression analysis using combined data from the cohorts of patients on ART and LTNPs. The Recent SRCV cohort was excluded from this analysis because of its high level of active replication, which would have biased the HIV-1 usRNA, 2-LTR, and total HIV-1 DNA measurements. For this analysis, patient-derived samples with detectable markers were included. This was confirmed by Spearman's rank correlation, which includes all samples (data not shown).

We found a positive correlation between HIV-1 usRNA and both total HIV-1 DNA (R$^2$ = 0.19, p < 0.001; *Figure 4B*) and integrated HIV-1 DNA (R$^2$ = 0.18, p < 0.01; *Figure 4C*). Total HIV-1 DNA correlated with integrated HIV-1 DNA levels (R$^2$ = 0.31, p < 0.001; *Figure 4D*). No correlation was observed between 2-LTR circles and HIV-1 usRNA, total or integrated HIV-1 DNA (data not shown).

A negative correlation was found between the CD4/CD8 ratio and integrated HIV-1 DNA (R$^2$ = 0.14, p < 0.01; *Figure 4E*). No correlation was found between the CD4/CD8 ratio and the other virological markers assessed (total HIV-1 DNA, 2-LTR circles and usRNA in PBMCs and total HIV-1 DNA in rectal biopsies), (data not shown).

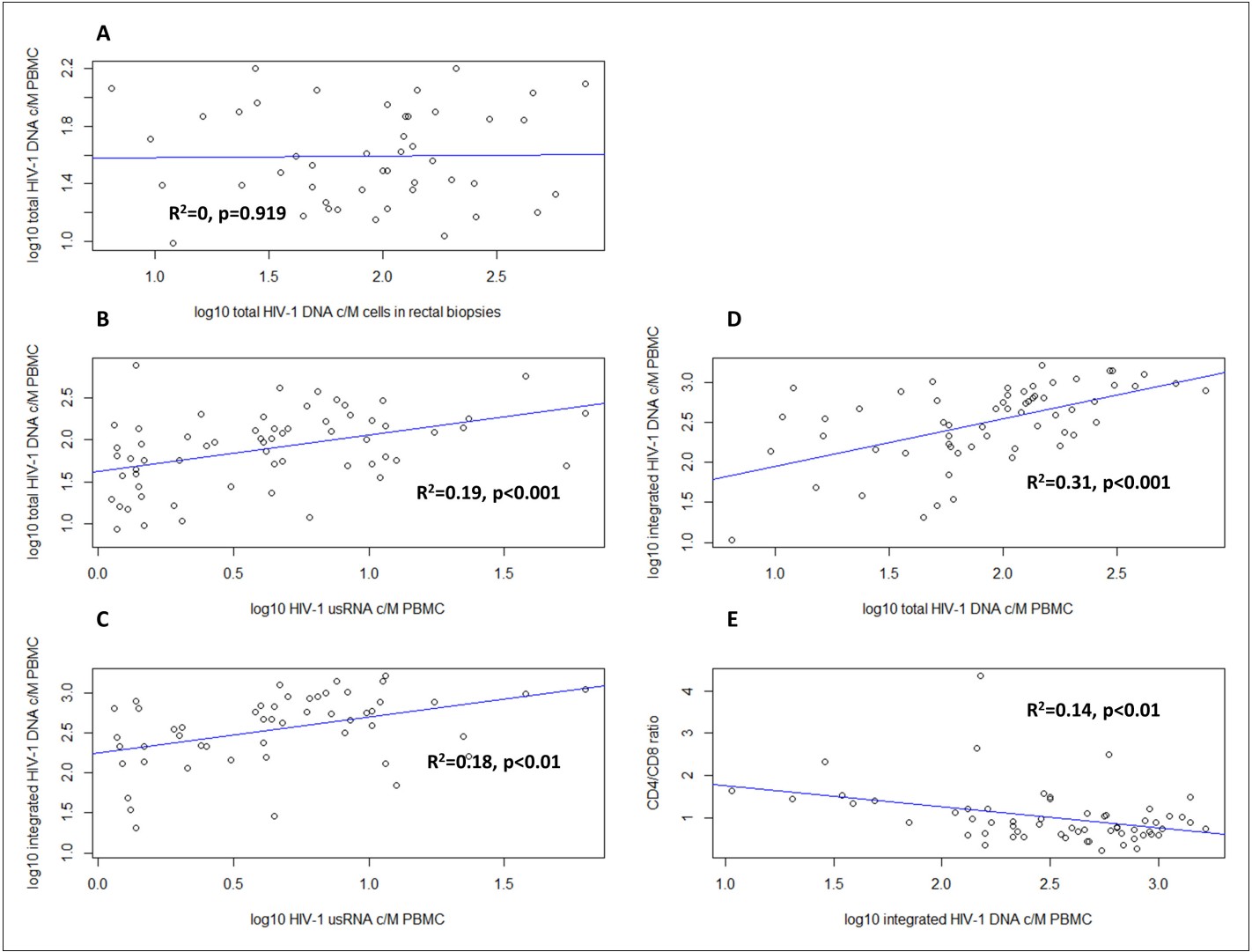

**Figure 4.** Correlation of total HIV-1 DNA in rectal biopsies and blood (**A**). Correlation was assessed in 46 patients in whom total HIV-1 DNA was detected both in the blood and rectal biopsies, representing patients from three cohorts: SRCV on antiretroviral therapy (ART), Chronic ART, and long-term non-progressors (LTNP). Data is shown as $\log_{10}$ copies/million (c/M) cells in rectal biopsies and $\log_{10}$ c/M peripheral blood mononuclear cells (PBMCs) in blood. Correlation of blood HIV-1 usRNA and total HIV-1 DNA (**B**), HIV-1 usRNA and integrated HIV-1 DNA (**C**), integrated HIV-1 DNA and total HIV-1 DNA (**D**) and CD4/CD8 ratio and integrated HIV-1 DNA (**E**). Data is shown as $\log_{10}$ c/M PBMCs in detectable patients from three cohorts: SRCV on ART, Chronic ART, and LTNP (**B–E**). To assess correlations between the markers, a linear regression was performed.

## Discussion

In the present study, we have shown that a decade of ART, initiated during seroconversion, decreases the HIV-1 DNA reservoir size and viral transcription level in blood, and benefits immunological restoration in comparison to later treatment initiation of the same duration. These data support the notion that early treatment initiation at the time of seroconversion may favor post-treatment viral control by limiting the establishment of an extended viral reservoir. However, we have also found that seeding of the viral reservoir occurs very rapidly after acquisition of the infection and, indeed, our data show that the small total and integrated HIV-1 DNA reservoir in early treated patients is still significantly larger despite early treatment initiation when compared to LTNPs.

As the majority of viral reservoirs is located in tissues, we sampled rectal tissues to assess the reservoir size in the gut mucosa. Our results do not show a higher total HIV-1 DNA levels in this compartment in both early and late ART-treated patients than in LTNPs. We did not find a correlation

between the blood and gut reservoirs. This is in accordance with one previous study (*Di Stefano et al., 2001*), but not with others (*Anton et al., 2003*; *Avettand-Fenoel et al., 2008*). A less efficient clearance of total HIV-1 DNA in rectal tissues compared to PBMCs has previously been described after early therapy initiation (*Chun et al., 2008*; *Ananworanich et al., 2012*). Low level cryptic HIV-1 replication may be a result of sub-optimal drug penetration in this compartment, immunological impairment, or caused by another unknown mechanism. Our results do not confirm such a hypothesis, although we have observed a non-significant trend towards a lower HIV-1 DNA level in gut in LTNPs compared to ART-treated patients. Lower levels of total HIV-1 DNA in gut were also previously detected in elite controllers compared to ART-suppressed patients (*Hatano et al., 2013b*). It remains unclear how the HIV-1 reservoir in the gut compartment might be influenced by prolonged early therapy initiation, as we did not have longitudinal samples, and whether it may contribute to viral rebound after therapy interruption. Furthermore, sampling the gut mucosa remains an invasive procedure compared to blood sampling, it cannot be performed as frequently and standardization to the exact sampling location is difficult.

Residual viral replication is a topic of intense debate in the HIV-1 cure field. Evidence is accumulating that the reservoirs may consist of truly latent provirus maintained in long-lived quiescent CD4 memory T cells, seeded soon after acute infection or maintained through homeostatic proliferation (*Maldarelli et al., 2014*; *Wagner et al., 2014*; *Cohn et al., 2015*). Low level residual viral replication may also represent an important mechanism in sustaining a replication-competent viral reservoir (*Sharkey et al., 2005*; *2011*; *2013*; *Hong and Mellors, 2015*). The presence of 2-LTR circles is considered indicative of viral replication and, as expected in this study, its levels were highest in recent ART-naïve seroconverters who display a high amount of ongoing replication. Yet no difference was observed between ART-treated patients and LTNPs. This suggests that ART effectively suppresses viral replication, regardless of the timing of treatment initiation. However, the value of 2-LTR circles as a marker for replication remains unclear as some reports suggest that 2-LTR circles are stable and long-lived (*Brussel et al., 2003*; *Pierson et al., 2002*). Notably, the amount of 2-LTR circles was low in a substantial number of ART-treated patients and LTNPs and undetectable in 26% of treated patients. Levels of 2-LTR circles did not correlate with total or integrated HIV-1 DNA or HIV-1 usRNA, indicating that factors other than the size of the viral reservoir may determine the presence of 2-LTR circles.

We have used cell-associated HIV-1 usRNA to reflect proviral DNA transcription. This marker may predict the replicative-competence of the viral reservoir, and previous studies have shown a correlation of usRNA levels with virological failure (*Pasternak et al., 2013*) and markers of immune activation in elite controllers (*Hunt et al., 2011*). In addition, recent trials using histone deacetylase inhibitors to stimulate viral production in the reservoir have used HIV-1 usRNA as a marker to assess viral transcription (*Archin et al., 2012*; *Elliott et al., 2014*; *Rasmussen et al., 2014*; *Wei et al. 2014*). An earlier report has shown that early-treated patients have lower HIV-1 usRNA compared to late-treated patients after a short period of ART (*Schmid et al., 2010*). With a long-term follow-up, Buzon et al. observed a trend towards lower levels of HIV-1 usRNA in elite controllers and early-treated patients compared to late-treated patients (*Buzon et al., 2014*). Here, we can confirm that early-treated patients have lower HIV-1 usRNA compared to patients who started therapy with onset of chronic HIV-1 infection. This observation persists after a decade of successful ART. The levels of HIV-1 usRNA in early-treated seroconverters were not different from those of LTNPs, suggesting that early treatment initiation reduces viral transcriptional activity to levels reminiscent of LTNPs, which may facilitate further intervention to control such low level of viral replication. However, it must be noted that HIV-1 usRNA levels positively correlated with those of total and integrated HIV-1 DNA. This may indicate that lower levels of HIV-1 usRNA are a consequence of a smaller pool of HIV-1 DNA reservoir rather than a low transcriptional activity. Cell-associated HIV-1 usRNA levels were shown to be very low in LTNPs, thus supporting its use as a reference parameter in future studies aimed at an HIV-1 cure.

T cell activation often remains elevated in chronic HIV-1 infection in spite of ART and is linked to a lower rate of CD4 T cell count recovery (*Goicoechea et al., 2006*) and higher mortality rates (*Serrano-Villar et al., 2014*). Lower levels of T cell activation were shown recently in early-treated patients within 6 months of infection with a shorter ART duration (median: 2.8 years) compared to later ART initiation (*Jain et al., 2013*). Here, we show that a higher CD4/CD8 ratio is found in long-term early-treated seroconverters with levels comparable to those observed in LTNPs, confirming

observations from a previous study (*Cellerai et al., 2011*). We were also able to show that the increased CD4/CD8 ratio was associated with a smaller integrated HIV-1 DNA reservoir, but not with total HIV-1 DNA. These two markers should therefore be further studied for their accuracy in reflecting the state of the HIV-1 reservoir in cure studies.

We have shown that the virological markers used in this study are affected by early treatment initiation. They could potentially represent predictors of a functional cure or of the time to viral rebound after ART interruption, where treatment interruption studies are necessary to validate these predictors. Although, the inter-patient variability of results may prevent the use of reservoir markers on an individual basis, a longitudinal follow-up of these markers may increase the rationale for their use in such studies. Recently, an association with time to viral rebound and HIV-1 DNA was found in early-treated patients within the SPARTAC study of seroconverters (*Williams et al., 2014*), and with integrated HIV-1 DNA in a study using pegylated Interferon alfa-2a to purge the viral reservoir (*Azzoni et al., 2013*). Hence, future studies should emphasize the longitudinal profiling of HIV-1 reservoir markers to validate their use in cure studies.

Some of the limitations of our study include the lack of pre-treatment and longitudinal follow-up in terms of viral reservoirs, which could provide information on their dynamics over the long-term and indicate whether a shorter period of treatment could achieve the same reservoir levels in both blood and tissues. The present study used PCR-based HIV-1 markers, which cannot differentiate between the presence or absence of replicative-competent reservoirs, and we did not use ex vivo assays such as the viral outgrowth assay (VOA) in blood, which can quantify replication-competent proviral HIV-1 DNA, the fraction of the reservoir that matters in terms of viral eradication. Both types of methods either overestimate or underestimate the size of the replicative-competent reservoir (*Bruner et al., 2015*), and VOA remains a cumbersome and expensive assay that is not widely available and will need to be replaced by easier and cheaper assays.

In conclusion, our study provides important data on the blood HIV-1 reservoir size and dynamics after a decade of successful treatment with ART together with gut mucosal HIV-1 DNA. Our results support early treatment initiation in terms of achieving lower levels of viral reservoirs, when compared to later treatment. The levels of HIV-1 DNA that are likely to be associated with a functional cure remain to be determined together with other factors such as protective immunological responses involved in virological control.

## Material and methods

### Patient cohorts

Eighty-four HIV-1 infected patients from four pre-defined cohorts under clinical follow-up were enrolled into the study in two clinical centers (The Ian Charleson Day Centre, Royal Free Hospital, London, United Kingdom and the AIDS Reference Center, Ghent University Hospital, Ghent, Belgium): long-term-treated patients with ART initiated during seroconversion or chronic infection, LTNPs, and recent ART-naïve seroconverters. The first three cohorts were recruited using databases at the clinical centers (the Royal Free Center Research Database and Ghent University Hospital Database). The fourth cohort consisting of acute seroconverters was enrolled prospectively in both clinical centers. The study was approved by the Ethics Committee of Ghent University Hospital (Reference number: B670201317826) and the Royal Free Hospital (Reference number: 13/LO/0729).

The patient cohorts are described in *Figure 1*. The first cohort consisted of HIV-1 seroconverters on long-term ART initiated and uninterrupted since PHI (SRCV on ART; n = 25). These patients were selected on the basis of the following inclusion criteria: (a) ART for $\geq$4 years; (b) long-term aviremia (<50 HIV-1 RNA copies (c)/ml); and (c) an absence of treatment failure defined by a viral load (VL) $\geq$400 HIV-1 RNA c/ml. Seroconversion to HIV-1 was defined by: (a) negative HIV-1 antibody by enzyme-linked immunosorbent assay (ELISA) and evidence of HIV-1 viremia $\geq$5000 HIV-1 RNA c/ml plasma and/or (b) incomplete HIV-1 Western blot with $\leq$3 bands and/or a detuned assay with a value of $\leq$0.6 for HIV-1 clade B patients.

The second cohort of LTNPs (n = 17) were therapy-naïve patients, with $\geq$7 years of documented HIV-1 infection, viremia $\leq$1000 HIV-1 c/ml and a CD4 T cell count $\geq$500 cells/mm$^3$ during follow-up. Exception was made for temporary ART to prevent mother-to-child transmission.

The third cohort consisted of HIV-1 infected patients successfully treated by ART initiated during chronic infection (Chronic ART; n = 32) with: (a) a treatment duration of ≥4 years and (b) long-term undetectable VL (<50 HIV-1 c/ml for ≥4 years). ART failure during follow-up was defined by a VL ≥400 HIV-1 c/ml.

The fourth cohort consisted of recent ART-naïve seroconverters (Recent SRCV; n = 10). Criteria for the diagnosis of seroconversion and enrollment into the study were the following: (a) negative or indeterminate HIV-1 antibody result by fourth generation ELISA and evidence of HIV-1 viremia ≥5000 HIV-1 c/ml plasma and/or (b) positive HIV-1 by fourth generation ELISA and two negative confirmatory tests (negative Vidas and Immunocomb) and/or (c) positive HIV-1 by fourth generation ELISA and negative InnoLIA.

Baseline characteristics, clinical and laboratory parameters including total duration of ART, VL zenith, CD4 T cell count at blood sampling and nadir, as well as CD4/CD8 ratio at blood sampling were collected and are summarized in *Table 1*.

Both the SRCV on ART and Chronic ART cohorts had a comparable uninterrupted ART duration with a median of 10.8 (IQR: 4.2–11.9) and 9.8 years (4.9–14.7), (p = 0.936), respectively.

## Blood and rectal biopsies

PBMCs and rectal biopsies were collected on one occasion for each patient (*Figure 1*). Blood was drawn in 6 EDTA 9 ml tubes and 10 rectal biopsies were sampled after obtaining written informed consent from the patients.

PBMCs were isolated within 4 hr of blood sampling by using Lymphoprep centrifugation (ELITech Group, Zottegem, Belgium). Cells were manually counted using a hemocytometer counting grid, aliquoted in $10^6$PBMCs as dry pellets or in fetal calf serum + 7.5% dimethyl sulfoxide and stored at −80°C. Flexible sigmoidoscopy was performed and 10 gut biopsies (volume around 1 mm$^3$ of each biopt) were taken from the mucosa of the upper rectum (10 cm from the anal margin) using single-use biopsy forceps. Intact rectal biopsies were immediately frozen and stored at −80°C until further processing.

PBMCs were collected from all patients (n = 84) included in the study and rectal biopsies were obtained from 51 patients, 14 in SRCV on ART, 8 in LTNP, and 29 in Chronic ART cohorts.

## Quantification of the HIV-1 reservoir, ongoing replication and transcription

Methods used to quantify virological parameters relating to HIV-1 reservoirs, ongoing replication, and transcription in this study have been recently published and include total HIV-1 DNA and 2-LTR circles (*Malatinkova et al., 2014*), cell-associated HIV-1 usRNA (*Kiselinova et al., 2014*) with the use of droplet digital PCR (ddPCR) and integrated HIV-1 DNA measured by a repetitive sampling Alu-HIV PCR (*Liszewski et al., 2009*; *De Spiegelaere et al., 2014*).

Total HIV-1 DNA, 2-LTR circles, and HIV-1 usRNA were measured in triplicates on ddPCR with the QX100Droplet Digital PCR platform (Bio-Rad, Hercules, CA). The ddPCR mix was made by adding 2 μl of sample (restricted genomic DNA [gDNA] or plasmid DNA) or 4 μl of sample (cDNA) to 10 μl 2× ddPCR™ supermix for probes (Bio-Rad), 500 nM of primers and 250 nM of probe in a final volume of 20 μl. ddPCR amplification reactions consisted of initial denaturation at 95°C for 5 min, followed by 40 cycles of 95°C for 30 s denaturation and assay-specific annealing/elongation temperature (*Supplementary file 1*) for 60 s with a ramp rate of 2.5°C/s. Droplets were read by the QX100 droplet reader (Bio-Rad) and the data was analyzed with the QuantaSoft analysis software (Bio-Rad). Primers and probes used for each quantification assay are summarized in *Supplementary file 1* and were purchased from IDT DNA Technologies (Integrated DNA Technologies, Leuven, Belgium).

## Total and integrated HIV-1 DNA

Total gDNA was isolated from $10^6$ PBMCs using DNeasy Blood & Tissue Kit (Qiagen, Venlo, The Netherlands) and eluted in 75 μl elution buffer, kept at 56°C for 10 min in order to maximize the DNA yield.

Three intact rectal biopsies were pooled per patient and used to isolate gDNA using DNeasy Blood & Tissue Kit (Qiagen). Total gDNA was eluted in 40 μl elution buffer to concentrate the sample and kept at 56°C for 10 min.

To measure total HIV-1 DNA, an enzyme restriction digestion with EcoRI (Promega, Leiden, The Netherlands) was performed on gDNA from both PBMCs and rectal biopsies with the use of 17.3 µl gDNA in a total volume of 20 µl of restriction mix. This step is preferred for ddPCR as the fragmented DNA will be more uniformly distributed in all droplets compared to full-length chromosomal DNA.

To quantify integrated HIV-1 DNA, gDNA isolated from PBMCs was subjected to a repetitive sampling Alu-HIV PCR, according to a recently described protocol (*De Spiegelaere et al., 2014*). Briefly, a method based on Poisson statistics was used to analyze the binomial data of positive and negative reactions from a 40-replicate Alu-HIV PCR (*De Spiegelaere et al., 2014*). First, total HIV-1 DNA was measured in gDNA samples by ddPCR and based on these measures, each sample was diluted to approximately five copies of total HIV-1 DNA per PCR replicate and distributed in replicates. Alu-HIV PCR has been described previously (*Liszewski et al., 2009*; *Yu et al., 2008*; *De Spiegelaere et al., 2014*) and was performed by using an HIV-1-specific reverse primer in *Gag* and a human *Alu*-specific forward primer (*Supplementary file 1*) in 40 replicates. In parallel, 20 replicates were run for background quantification by using only the HIV-1 *Gag* primer. The PCR mix was made by adding 10 µl of diluted gDNA sample to 10 µl PCR mix consisting of $5\times$ Go Taq G2 master mix, 0.2 µl Go Taq G2 DNA Polymerase, 4 mM of dNTP mix (Promega), 200 nM of *Alu* primer and 1200 nM of HIV-1 primer in a final volume of 20 µl. PCR amplification reactions consisted of initial denaturation at 95°C for 2 min, followed by 40 cycles of 95°C for 15 s denaturation, 50°C for 15 s annealing, and 70°C for 3.5 min elongation. 2 µl of the PCR product were processed in the nested qPCR (Light Cycler 480 System, Roche Applied Science, Penzberg, Germany), qPCR mix contained $2\times$ LightCycler 480 Probes Master mix (Roche Applied Science, Vilvoorde, Belgium), 400 nM primers, and 200 nM probe (*Supplementary file 1*), and qPCR consisted of initial denaturation at 95°C for 5 min, followed by 45 cycles of 95°C for 15 s denaturation, and 60°C for 1 min annealing/elongation.

The quantities of total and integrated HIV-1 DNA c/$10^6$ PBMCs or total HIV-1 DNA c/$10^6$ cells in rectal biopsies were normalized to a reference gene *RPP30* measured by ddPCR. The number of cells assayed per patient was measured by *RPP30* in all PBMCs and rectal biopsies samples.

## Episomal HIV-1 2-LTR circles

Episomal HIV-1 2-LTR circles were measured in plasmid DNA isolated by QIAprep Spin Miniprep kit (Qiagen) from dry pelleted $10^6$ PBMCs. A known number of pSIF1-H1-Puro non-HIV plasmid was spiked to the samples (System Biosciences, Mountain view, CA) as an internal control for copy number normalization and plasmid DNA was eluted in 25 µl in order to increase DNA concentration, as described previously (*Malatinkova et al., 2014*). The internal reference plasmid was quantified by detection of the woodchuck hepatitis virus posttranscriptional regulatory element (WPRE) (*Lizee et al., 2003*). The 2-LTR assay is designed to span over the 2-LTR junction (*Buzon et al., 2010*; *Supplementary file 1*).

## Cell-associated HIV-1 usRNA

RNA was isolated from $10^6$ PBMCs by using RNeasy mini kit (Qiagen) subjected to DNase treatment by RNase-Free DNase Set (Qiagen) and eluted in 30 µl nuclease-free water. Samples were measured by NanoDrop 2000 (Thermo Fisher Scientific, Waltham, MA) and 1.5 mg of RNA was processed by the iScript cDNA Synthesis Kit (Bio-Rad) 5 min at 25°C, 30 min at 42°C, and 5 min at 85°C. The cDNA was used to measure HIV-1 usRNA on ddPCR.

Normalization of input cDNA was performed by quantifying gene expression of stably expressed internal reference genes as described earlier (*Ceelen et al., 2014*; *Messiaen et al., 2012*). Briefly, the three most stably expressed reference genes (from total of nine genes tested) were selected over all patient samples by geNorm analysis (Beta-2-microglobulin: B2M, TATA box binding protein: TBP, and Ubiquitin C: UBC) (*Vandesompele et al., 2002*). Normalization factors were determined per patient as the geometric mean of the three most stable reference genes. Subsequently, raw ddPCR values for HIV-1 usRNA were divided by the normalization factors to reach normalized data and reported as c/$10^6$ PBMCs for each patient sample. Previously described primers and probe sets for HIV-1 usRNA quantification were used (*Kiselinova et al., 2014*; *Palmer et al., 2008*), as summarized in *Supplementary file 1*.

## Statistical analysis

Total HIV-1 DNA, integrated HIV-1 DNA, 2-LTR circles and cell-associated HIV-1 usRNA levels as well as immunological data (CD4/CD8 T cell ratios and CD4 T cell counts at sampling and nadir CD4 T cell counts) were described using median values and IQR. Statistical analysis was performed using R (RStudio, Inc., Boston, MA). Standard non-parametric test (Wilcoxon Signed Rank Test) was performed to assess statistically significant differences between patient cohorts. A p-value of <0.05 was considered significant. Linear regression was used to assess the correlations.

## Acknowledgement

This work was performed by the support of The Foundation for AIDS Research (AmfAR) (Grant ID: 108314-51-RGRL), without pharmaceutical sponsoring. EM and PB are supported by the Agency for Innovation by Science and Technology of the Flemish Government (IWT, Grant nr: 111286 and 111393). MK is supported by a 'Special Research Grant – BOF grant' of Ghent University (Grant nr: 01N02712). LV is a Senior Clinical Investigator supported by FWO (Grant nr: 1.8.020.09. N.00). Additionally, this work was supported by the King Baudouin Foundation (Grant nr: 2010-R20640-003) and HIV-ERA (130442 SBO, EURECA).

## Additional information

### Funding

| Funder | Grant reference number | Author |
|---|---|---|
| amfAR, The Foundation for AIDS Research | Grant ID: 108314-51-RGRL | Linos Vandekerckhove |
| Agentschap voor Innovatie door Wetenschap en Technologie | PhD Fellowship, Grant nr: 111286 and 111393 | Eva Malatinkova Pawel Bonczkowski |
| Special Research Grant- BOF grant of Ghent University | PhD Fellowship, Grant nr: 01N02712 | Maja Kiselinova |
| Fonds Wetenschappelijk Onderzoek | Senior Clinical Investigator, Grant nr: 1.8.020.09.N.00 | Linos Vandekerckhove |
| Koning Boudewijnstichting | Grant nr: 2010-R20640-003 | Linos Vandekerckhove |
| HIV-ERA | 130442 SBO, EURECA | Linos Vandekerckhove |

The funders had no role in study design, data collection and interpretation, or the decision to submit the work for publication.

### Author contributions

EM, Acquisition of data, Analysis and interpretation of data, Drafting or revising the article, Contributed unpublished essential data or reagents; WDS, MK, Analysis and interpretation of data, Drafting or revising the article, Contributed unpublished essential data or reagents; PB, KV, WT, CV, Acquisition of data, Drafting or revising the article, Contributed unpublished essential data or reagents; MJ, DdeL, CM, Conception and design, Acquisition of data, Drafting or revising the article; SKdeL, LV, Conception and design, Acquisition of data, Analysis and interpretation of data, Drafting or revising the article

### Author ORCIDs

Ward De Spiegelaere, http://orcid.org/0000-0003-2097-8439
Linos Vandekerckhove, http://orcid.org/0000-0002-8600-1631

### Ethics

Human subjects: Patient written informed consent was obtained from all the study participants. The study was approved by the Ethical Committee of Ghent University Hospital (Reference number: B670201317826) and Royal Free Hospital (Reference number: 13/LO/0729).

## Additional files

**Supplementary files**
• Supplementary file 1: Summary of primers/probe sets for PCR-based virological quantification.

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
