## [Decision Letter]

Thank you for submitting your work entitled "Impact of a decade of successful antiretroviral therapy initiated at HIV-1 seroconversion on blood and mucosal reservoir" for peer review at *eLife*. Your submission has been favorably evaluated by Prabhat Jha (Senior Editor), a Reviewing Editor, and two peer reviewers, one of whom, Camilla Tincati, has agreed to reveal her identity.

The reviewers have discussed the reviews with one another and the Reviewing editor has drafted this decision to help you prepare a revised submission.

This paper on HIV reservoirs contributes to advancing the HIV cure agenda. Investigations of blood and rectal tissue in cART-treated subjects who started therapy either at time of HIV acquisition or during chronic infection were compared. Study patients were additionally compared to long-term non progressors and untreated seroconverters, representing, respectively a paradigm of virological control and early stage viremia. Initiation of cART early in the course of disease reduces total and integrated HIV DNA, yet not to the levels observed in LTNP. In contrast, timing of cART seems to affect the kinetics of usRNA. HIV reservoirs in the rectal mucosa in treated subjects were comparable to those observed in LTNP, regardless of the timing of treatment initiation. The data suggest that early, long-term cART has an important, but limited role in restraining HIV reservoirs. The novelty of this investigation, notwithstanding small numbers, lies in its attempt to perform a comprehensive analysis of HIV reservoirs in subjects starting cART in different stages of disease.

Essential revisions:

1) Introduction:

1.1) The authors refer to "functional cure" several times. This term should be defined early in the Introduction.

1.2) When discussing functional cure, some clarifications are needed. For example, "LNTPs represent a paradigm of functional cure" is an overstatement. The definition of LTNPs in this study is VL <1000 copies/ml but the working definition for functional cure is a more tightly controlled viral load (<50 copies/ml). Even achieving elite control status does not necessarily equate to functional cure (because of persistent immune activation in EC which is not a desired goal of functional cure). The authors need to be more cautious about equating LTNPs to functional cure.

1.3) The Introduction is very lengthy and some parts overlap with the Discussion. Partially introducing the role of CD8 + T cells in functional cure is confusing and maybe misleading to the readers. This is because the example of functional cure cases – the VISCONTI patients – did not have a strong CD8 + T cell response. Moreover, the authors are not presenting HIV-specific CD8 responses data in this manuscript.

2) Methods:

2.1) For the rectal biopsy, please provide more details about the tissue obtained for reservoir assessment: size of tissue pieces (or size of forcep), number of pieces, and number of cells assayed.

3) Results:

3.1) 2LTR would be elevated if integrase inhibitors were used. Please clarify if this drug class was used in any of the treated groups.

4) Discussion:

4.1) The Discussion section is very lengthy and should be more concise. For instance, the long discussion about the START trial could be shortened to just mentioning the significance of the results giving momentum to earlier treatment initiation.

4.2) The authors mentioned that 2LTR circles is a labile product and marker of ongoing viral replication, however, there have been data suggesting that 2LTR circles are stable and long-lived (12; 51). This may explain the lack of differences between the treated groups.

4.3) Another limitation that should be mentioned is the sampling limitation of gut biopsy in terms of frequency of cells and sampling area.

4.4) The authors' conclusion that the inability to reduce reservoir size in treated people to the level seen in LTNP means that additional interventions will be needed for functional cure is problematic. Although there is agreement that combination interventions will likely be needed for functional cure, the authors are attributing sole importance to the reservoir size as a marker of functional cure. We know that much of the total DNA is defective and that the VISCONTI patients actually had a readily detectable DNA (2 log) so there is much we do not know.

There remains concerns about using LTNP as a "gold standard" and the authors should reflect in the Discussion and conclusion based on their data 1) earlier treatment is superior to later treatment in reducing reservoir size, 2) LTNP is used as an example of the ability to moderately control viremia (not functional cure) in the absence of ART, 3) the levels of DNA that are important to achieving functional cure are unknown and many factors are involved. Evidence so far suggests that combination therapy will likely be needed to achieve functional cure.

---

## [Author Response]

Essential revisions:

1) Introduction:

1.1) The authors refer to "functional cure" several times. This term should be defined early in the Introduction. 

In the revised manuscript, the meaning of “functional cure” has been clarified early in the Introduction as “virological control in the absence of therapy”.

1.2) When discussing functional cure, some clarifications are needed. For example, "LNTPs represent a paradigm of functional cure" is an overstatement. The definition of LTNPs in this study is VL < 1000 copies/ml but the working definition for functional cure is a more tightly controlled viral load (< 50 copies/ml). Even achieving elite control status does not necessarily equate to functional cure (because of persistent immune activation in EC which is not a desired goal of functional cure). The authors need to be more cautious about equating LTNPs to functional cure. 

We agree with the reviewers’ comment about the absence of similarity between the functional cure paradigm and LTNPs. In the new version of the manuscript, this statement has been removed.

1.3) The Introduction is very lengthy and some parts overlap with the Discussion. Partially introducing the role of CD8 + T cells in functional cure is confusing and maybe misleading to the readers. This is because the example of functional cure cases – the VISCONTI patients – did not have a strong CD8 + T cell response. Moreover, the authors are not presenting HIV-specific CD8 responses data in this manuscript. 

To avoid any confusion and because indeed we do not present immunologic data in our study, the information on the role of CD8 + T cells in functional cure has been removed. We have also made the Introduction shorter and tried to avoid an overlap with the Discussion section.

2) Methods:

2.1) For the rectal biopsy, please provide more details about the tissue obtained for reservoir assessment: size of tissue pieces (or size of forcep), number of pieces, and number of cells assayed. 

Fifty-one patients had agreed to a flexible sigmoidoscopy and 10 rectal biopsies were taken from each patient with a volume of around 1 mm^3^ for each biopsy. Biopsies were taken from the mucosa of the upper rectum (10 cm from the anal margin) using single-use biopsy forceps (subsection “Blood and rectal biopsies”). The number of cells assayed per patient was added to the Results section (subsection “Total HIV-1 DNA levels in rectal biopsies are not different in early treated seroconverters, late treated patients and LTNPs”).

3) Results:

3.1) 2LTR would be elevated if integrase inhibitors were used. Please clarify if this drug class was used in any of the treated groups. 

Some of the patients were receiving an integrase inhibitor (n = 9), but no difference in 2-LTR levels was observed when these were compared to the other ART-treated patients. There are several reports that have highlighted a transient rise in 2-LTR levels after a treatment switch to an integrase inhibitor. However, both Buzon et al. and Hatano et al. have noted that this rise is transient as the 2-LTR increase is most pronounced at week 2 after the switch to an integrase inhibitor, but decreases in the ensuing weeks. In our patients, we did not observe a difference in 2-LTR levels, which would support previous findings, as they had been treated over the long-term without a recent treatment switch in the weeks before sampling.

4) Discussion:

4.1) The Discussion section is very lengthy and should be more concise. For instance, the long discussion about the START trial could be shortened to just mentioning the significance of the results giving momentum to earlier treatment initiation. 

As suggested by the reviewers, the Discussion has been shortened. We hope that these changes are satisfactory.

4.2) The authors mentioned that 2LTR circles is a labile product and marker of ongoing viral replication, however, there have been data suggesting that 2LTR circles are stable and long-lived (12; 51). This may explain the lack of differences between the treated groups. 

We are aware of the current debate on 2-LTR circles as a marker of viral replication. In the revised manuscript we have highlighted the current controversy on the use of this marker (Discussion, third paragraph) and hope to have clarified the issue.

4.3) Another limitation that should be mentioned is the sampling limitation of gut biopsy in terms of frequency of cells and sampling area. 

There are indeed practical and potentially ethical considerations in performing biopsies in patients such as LTNPs and in terms of a more invasive type of biopsy sampling in the sigmoid or small intestine. Our choice of the rectal area was driven by the fact that we had an excellent uptake by patients in another study and that it seems to be minimally invasive as compared to a more comprehensive biopsy protocol and could potentially be used in cure studies with minimal patient discomfort. By using 10 samples per patient we believe that we were be able to obtain a sufficient number of cells to detect and quantify HIV-1 DNA.

To highlight the practical issues, we have added a small section to the Discussion of the revised manuscript. These are indeed practical limitations that prevent the standardization of such sampling and hamper frequent monitoring. This has been added in the Discussion section. Notably, rectal sampling remains an invasive procedure as compared to blood sampling, which can be performed more frequently.

4.4) The authors' conclusion that the inability to reduce reservoir size in treated people to the level seen in LTNP means that additional interventions will be needed for functional cure is problematic. Although there is agreement that combination interventions will likely be needed for functional cure, the authors are attributing sole importance to the reservoir size as a marker of functional cure. We know that much of the total DNA is defective and that the VISCONTI patients actually had a readily detectable DNA (2 log) so there is much we do not know.

There remains concerns about using LTNP as a "gold standard" and the authors should reflect in the Discussion and conclusion based on their data 1) earlier treatment is superior to later treatment in reducing reservoir size, 2) LTNP is used as an example of the ability to moderately control viremia (not functional cure) in the absence of ART, 3) the levels of DNA that are important to achieving functional cure are unknown and many factors are involved. Evidence so far suggests that combination therapy will likely be needed to achieve functional cure. 

We absolutely agree with these comments and have attempted to make the necessary changes to the Discussion.

In our study, we have explicitly looked at various markers of the HIV-1 reservoir size and dynamics and compared results between our four patient populations. Our emphasis on reservoirs stems from the fact that data relating to HIV-1 DNA and viral rebound are available and suggest that the magnitude of the reservoirs is indeed important in terms of viral rebound and functional cure (64). It is clear that a large proportion of this reservoir is not replicative competent however no absolute measurement of this reservoir component is available and we have therefore tried to improve our understanding of the reservoir by using several virological parameters.

We have not performed in-depth characterization of any immunological aspects which may be driving virological control and made comparison between our patient cohorts. However, recent evidence indicates that the functional cure state achieved by some patients is most probably a result of the interplay between immunological and virological aspects, the former remaining still poorly understood due to the low numbers of post-treatment controllers, the difference in T cell responses between elite controllers and post-treatment controllers and an absence of studies analysing innate immunity.

We agree with the reviewers that LTNPs represent an imperfect surrogate group for functional cure, however, we believe that they represent some of the presently best available examples of viremia control with limited immune suppression over the long-term and this in the absence of a better comparator group of patients.

In order to make these points clear, we have adapted the conclusion section of the Discussion and hope that these changes have clarified pending issues.